# The methylation status of the embryonic limb skeletal progenitors determines their cell fate in chicken

Cristina Sanchez-Fernandez[1,2], Carlos Ignacio Lorda-Diez [1,2], Juan M. Hurlé [1✉] & Juan Antonio Montero [1✉]

Digits shape is sculpted by interdigital programmed cell death during limb development. Here, we show that DNA breakage in the periphery of 5-methylcytosine nuclei foci of interdigital precursors precedes cell death. These cells showed higher genome instability than the digit-forming precursors when exposed to X-ray irradiation or local bone morphogenetic protein (BMP) treatments. Regional but not global DNA methylation differences were found between both progenitors. DNA-Methyl-Transferases (DNMTs) including DNMT1, DNMT3B and, to a lesser extent, DNMT3A, exhibited well-defined expression patterns in regions destined to degenerate, as the interdigital tissue and the prospective joint regions. *Dnmt3b* functional experiments revealed an inverse regulation of cell death and cartilage differentiation, by transcriptional regulation of key genes including *Sox9*, *Scleraxis*, *p21* and *Bak1*, via differential methylation of CpG islands across their promoters. Our findings point to a regulation of cell death versus chondrogenesis of limb skeletal precursors based on epigenetic mechanisms.

[1] Departamento de Anatomía y Biología Celular and IDIVAL, Universidad de Cantabria, Santander 39011, Spain. [2] These authors contributed equally: Cristina Sanchez-Fernandez, Carlos Ignacio Lorda-Diez. ✉email: hurlej@unican.es; monteroja@unican.es

The formation of free digits in tetrapods takes place by a sculpting process that removes the interdigital tissue from the hand/foot of embryonic primordia. The process is reduced or abolished in vertebrate species with webbed digits (duck, bat, etc.) and its inhibition in species with free digits causes syndactyly. At the mechanistic level, interdigit remodeling utilizes numerous degenerative pathways including canonical apoptosis mediated by caspases, cell death associated with lysosomal activation, necrosis, autophagy, and cell senescence (reviewed by ref. [1]). Despite this mechanistic complexity, the process is considered to be one of the most illustrative examples of embryonic programmed cell death, and is assumed to represent a massive "cell suicide" process. However, at the cellular level, the degenerative process does not respond to what could be expected from a cell suicide process, as it is preceded by DNA breaks that cells try to repair[2]. In addition, we are still far from understanding how the different degradation pathways are coordinated, and whether they depend upon common or distinct regulatory signals. Importantly, no specific death signals, other than oxidative stress, have been identified in the embryonic limb[3–6]. Thus, while it is well established that BMPs induce cell death in interdigital progenitors[7], the same signals stimulate cell proliferation and differentiation in the tips of the fingers, which are located a few microns away from the cells that die[2]. Furthermore, the removed interdigital cells are skeletal progenitors that are able to form an extra digit under different experimental conditions[8].

Spontaneous DNA damage and the repair response have received much attention in studies of cancer[9,10] where breaking of DNA occurs after the disorganized accumulation of tumor stem cells and the transformation of the tissue architecture. It is likely that comparable features may occur in the embryonic degenerative processes. During digit formation, a critical transition between undifferentiated progenitors and stable differentiated cells occurs in the nascent digits but is not found in the interdigital regions. Remarkably, Smad2/3-TGFbeta signaling appears to be a crucial signal that directs progenitors to form digits in the presence of Smad1/5/8-BMP signaling[11]. Direct application of BMPs to embryonic chick interdigital progenitors causes intense DNA breaks that are followed by massive apoptosis and senescence[2]. However, if the interdigit cells are previously exposed to TGFbetas or Activins, they escape from the death program and form an extra digit[8].

It is now well known that environmental changes and growth factor signaling, cause epigenomic modifications that adjust gene expression by modifying the accessibility of the genes to transcription factors. DNA methylation and post-translational histone modifications are the most common epigenetic markers. It has been found that DNA methylation, is not static; rather, it is a dynamic cellular feature under the control of molecular signaling pathways[12]. Furthermore, we have recently found evidence for a role for UHRF multifunctional epigenetic regulators in the removal of the interdigital tissue via changes in global and regional methylation[13]. Therefore, an outstanding question in the understanding of embryonic sculpting processes is whether epigenetic mechanisms sensitize the skeletal progenitors to DNA damage signals that are harmless to cells committed to chondrogenesis.

Here, we show that prior to the onset of cell death, interdigital cells bear higher genome instability than the digit-forming precursors. This difference was associated with a differential expression of DNA-Methyl-Transferases in digit, joint primordia and interdigital tissues. Consistent with the pattern of gene expression, functional experiments in the micromass culture assay revealed an intense influence of DNA-Methyl-Transferase 3b promoting cell death and inhibiting chondrogenesis. These findings were associated with changes in the methylation status of the CpG islands of the promoter of key genes including *Sox9*, *Scleraxis*, *p21*, and *Bak1*.

## Results

**Differential DNA fragility to X-irradiation**. The sensitivity of the autopodial tissues to X-irradiation was monitored in embryos at incubation day 5.5 (id 5.5; HH stage 28), preceding the onset of physiological cell death by 24 h. We employed γH2AX immunolabeling as a marker of DNA damage and the TUNEL assays to detect apoptosis. γH2AX is formed by the phosphorylation at Serine 139 of the histone variant H2AX as a precocious response to DNA breaking; thus, γH2AX serves to identify DNA damage. TUNEL is a canonical marker of apoptosis that reflects major breaks of DNA that do not activate repair mechanisms.

Embryo survival was dependent on the radiation dose. Doses between 1 and 4 Gy were sublethal, and all embryos survived for more than 4 days ($n = 50$). Doses of 5 or 6 Gy, were lethal for 100% of the treated embryos ($n = 40$).

γH2AX labeling and TUNEL were induced in the autopods irradiated at sublethal doses when physiological cell death has not yet begun (Fig. 1a, b). Those degenerative marks were very abundant in the interdigital regions and subapical mesoderm, but they were almost absent from the digit rays (Fig. 1a, b). At doses of 3 and 4 Gy degeneration was higher, and degenerating cells extended to the margins of the digits and to the developing joints (Fig. 1d, f). The developing joint primordia show areas of physiological cell death, and, as shown in Fig. 1c, d, they become dramatically intensified by the irradiation.

At the cellular level, both γH2AX and TUNEL were detected 3 h after irradiation and their number increased in subsequent post-irradiation periods. The relative number of TUNEL-positive versus γH2AX cells increased with the doses of irradiation and, most important, with the time period elapsed after irradiation (Fig. 1e, g–i), suggesting that, at least in part, cells suffering DNA damage evolve to apoptosis. Indeed, by 12 h or more after irradiation most degenerating cells were TUNEL-positive (Fig. 1f, i).

Consistent with the absence of degenerating cells in the digit rays subjected to 1 or 2 Gy radiation, the autopodial skeleton was normal in all embryos subjected to 1($n = 15$) or 2 ($n = 15$) Gy (Supplementary Fig. 1a, b). Relatively mild skeletal alteration consisting of the absence of interphalangeal joints, and a reduction in the number of phalanxes was observed in 5 out of 10 experimental embryos irradiated with 3 Gy (Supplementary Fig. 1c). Exposure to 4 Gy caused joint inhibition and moderate (Supplementary Fig. 1d) or severe (Supplementary Fig. 1e) digit truncations in nine out of 10 experimental embryos. The pattern of senescence and cell death in malformed autopods at advanced stages of development revealed a strict inverse spatial correlation with chondrogenesis (Supplementary Fig. 1f), which suggests that cell differentiation and cell death are opposite and exclusive fates for the progenitors, similar to two sides of the same coin.

**BMPs promote interdigital DNA damage**. To further explore whether interdigital and digital cells show distinct susceptibility to DNA damage, we compared the rate of DNA damage induced in vivo by local application of BMPs in the interdigits and in the digit tips. BMP 2, 4, 5, and 7 are considered to be the signals that trigger interdigital cell death in physiological conditions[7]. As shown in Fig. 2a, b, 6 h after local application of BMP-beads (either BMP5 or BMP7) into the interdigits and in the neighboring digit tip resulted in the induction of a fully distinct pattern of γH2AX-positive cells. In the interdigits the BMP-bead induces massive DNA damage, while, none, or only few cells positive for γH2AX were induced by beads implanted at the digit tip. These

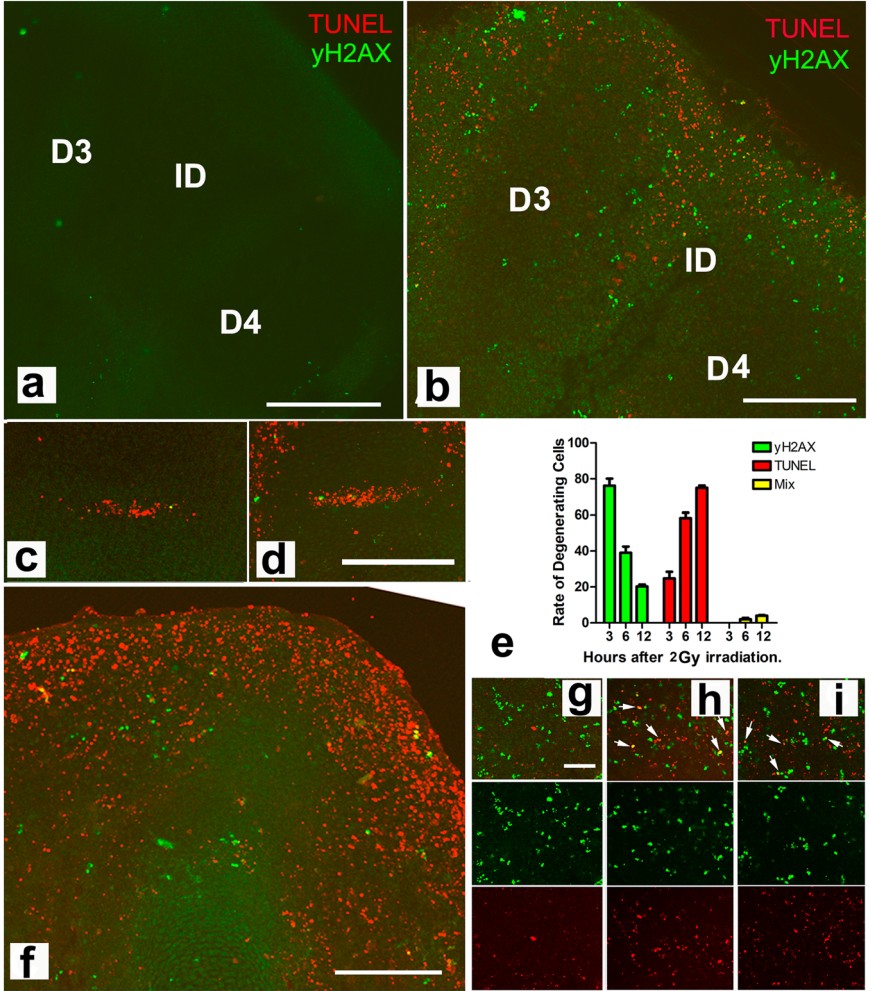

**Fig. 1 Degenerative events caused by X-irradiation. a, b** sections of control non-irradiated (**a**) and experimental autopod (**b**) 6 h after sublethal irradiation (3 Gy) at id 5 showing the predominant distribution of TUNEL-positive (red labeling) and γH2AX-positive cells (green immunolabeling) in the subapical and interdigital regions. Note the absence of degenerative marks in digits 3 (D3) and 4 (D4) flanking the third interdigit (ID). **c** and **d** are detailed views of a metatarsal-phalangeal joint interface in control (**c**) and irradiated (**d**) autopods showing the dramatic increase in TUNEL-positive cells (red labeling) 12 h after 4 Gy irradiation at id 5.5. **e** plot showing the percentage γH2AX-positive (green columns), TUNEL-positive (red); and γH2AX-TUNEL double-positive (yellow) in the third interdigit of chick limbs 3, 6, and 12 h after 2 Gy irradiation at id 5.5 (n = 4 biologically independent experiments). **f** section of a digit tip 12 h after 4 Gy irradiation at id 5.5 to show the intensity of apoptosis (red TUNEL labeling) in the subapical digit progenitors, but not in the prechondrogenic aggregate. **g–i** interdigit sections double labeled for TUNEL (red) and γH2AX (green), 3 (**g**), 6 (**h**), and 12 h (**i**) after 2 Gy irradiation (merged images: upper row; green channel: middle row; red channel: lower row). Note the progressive decrease in γH2AX-positive cells (green channel) and the increase in TUNEL-positive cells (red channel). Arrows show cells double-positive for TUNEL and γH2AX. Bars = 200 μm (**a**, **b**, **c**, **d**, **f**); 50 μm (**g–i**).

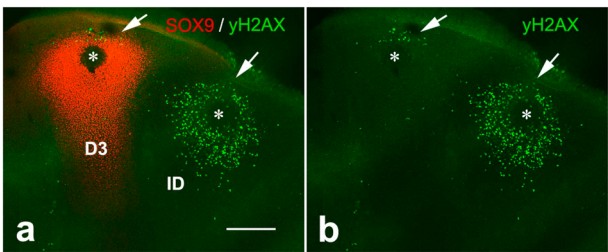

**Fig. 2 BMPs are specific DNA damaging signals for undifferentiated progenitors. a** confocal image of an experimental autopod 6 h after the implantation at id 5.5 of BMP5-beads (*) in the third interdigit (ID) and in the tip of digit 3 (D3) immunolabeled with anti- γH2AX (green) and anti-Sox9 (red); **b** shows only the green channel to better illustrate the regions with γH2AX-positive cells. Note the induction of a very intense area of cells positive γH2AX around the interdigital bead that contrast with a reduced number of cells γ H2AX-positive located distally to the digit bead (arrows). Bar = 200 μm.

cells were always located distally to the BMP-bead, and, most likely, represent still undifferentiated progenitors in the course of incorporation into the digit prechondrogenic aggregate. As previously reported, at longer time periods, cell death and chondrogenesis appeared intensified in the treated interdigits and digit tips, respectively (Supplementary Fig. 1g-i).

**DNA methylation of interdigital progenitors fated to die.** We investigated in normal limbs whether the high sensitivity to irradiation of the interdigital cells in the stages preceding physiological cell death, was associated with their methylation profile. To this end, we used an ELISA to assess global methylation values of interdigital and digit tip progenitors, prior to the appearance of signs of DNA damage (id 5.5), when DNA damage is first recognized (id 6), and, once the TUNEL-positive degenerative process is fully established in the interdigit regions (id. 7; Fig. 3a, b); (Supplementary Fig. 2 illustrates the normal course of interdigital tissue degeneration). As shown in Fig. 3c, differences in

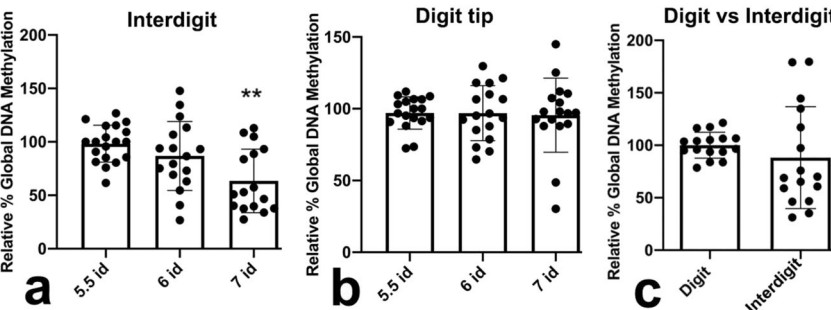

**Fig. 3 Global DNA methylation of interdigital and digit tip progenitors. a–c** charts showing the values of global methylation of the interdigit (**a**) and digit (**b**) progenitors at id 5.5, 6, and 7 evaluated by ELISA ($n = 16$ biologically independent samples). The value at id 5.5 was considered 100%. Note the decreased methylation at id 7 when the area of interdigital cell death is fully established. **c** comparative values of global methylation between the interdigital mesoderm and the digit tip mesoderm and id 6, to show absence of significant differences. $^{**}p < 0.01$.

global methylation between digit tips and interdigits were not detected prior to the establishment of the regression process.

The pattern of DNA methylation of the interdigits and digit progenitors was next monitored by immunofluorescence in dissociated cells using 5-methylcytosine (5mC) immunolabeling (Fig. 4a–f). During the whole period of tissue remodeling, the nuclei of digit and interdigital cells showed robust 5mC labeling in a characteristic dotted pattern (Fig. 4a–f). The 5mC marks showed variable sizes ranging from dots of less than 0.1 µm in diameter up to clumps of more than 0.5 µm. A mild decrease in the number of 5mC foci was detected in the interdigital cells by id 7.5 (Fig. 4d, e), when interdigital degeneration reaches peak level. The association of 5mC foci with markers of DNA damage at the stages of overt degeneration showed a preferential distribution of γH2AX that was in line with the methylated DNA (Fig. 4g–j). The analysis of the distribution of a total of 576 γH2AX foci from ten different samples of interdigital tissue at id 7.5, showed that 386 (67%) of the foci were located in or around the 5 mC foci and only 190 (33%), were far from them (more than 0.1 µm). This difference is more remarkable considering that the 5mC foci occupied less than 10% of the total surface of the nuclei. Furthermore, increased intensity of γH2AX labeling was associated with the coalescence of 5mC marks moving toward the center of the nuclei and forming large clumps of methylated DNA (Fig. 4k–m). The association between 5mC and γH2AX foci were less evident in progenitors subjected to irradiation. As shown in Fig. 4n–q, 12 h after 2Gy irradiation, γH2AX immunolabeling, that is negative in control un-irradiated progenitors (Fig. 4n, o), appeared as very small foci widespread through the nucleoplasm of irradiated progenitors, including the contour of the 5mC foci (Fig. 4p, q).

The combination of 5mC labeling with the TUNEL assay, showed that apoptosis is accompanied by peripheral displacement and progressive fading of 5mC labeling (Fig. 4r–t). This is consistent with the reported disorganization of the chromatin caused by phosphorylation of the high mobility group of chromatin remodeling factors (HMGA)[14], which occurs prior to the activation of DNase by caspases[15]. Other selected markers of DNA damage repair, such as MDC1, also showed a preferential distribution around the 5mC foci (Fig. 4u–x).

**Expression of methylation enzymes**. DNA methylation is carried out by three distinct DNA Methyl Transferases (DNMTs), and demethylation can occur passively during DNA replication or actively through enzymatic DNA demethylation[16]. DNMT1 is responsible for maintenance of methylation via transferring methyl groups to the hemi-methylated DNA strands following DNA replication. DNMT3A and DNMT3B cause de novo

methylation of cytosine residues (reviewed in ref. [17]). As shown in Fig. 5a, the interdigital expression of all Dnmt´s was very high when compared with HMGN1, which is a gene considered to be a good marker of undifferentiated interdigital mesoderm[18]. In fact, their PCR Ct values were similar to those detected in samples of dorsal neural tube and neural crest regions obtained from early embryos, where the occurrence of high expression levels of Dnmts was previously reported[19,20]. The highest expression level corresponded to Dnmt3b, and the lowest to Dnmt1; however, even in this case the expression was much higher than that of the selected interdigital gene marker (Fig. 5a). Analysis by in situ hybridization showed well-defined expression domains of Dnmt1 and Dnmt3b in the interdigital tissue and in the developing joint regions (Fig. 5b–d, i, j), although transcripts were present at lower levels in the remaining autopodial tissues. Dnmt3a was expressed more widely than Dnmt1 and Dnmt3b (Fig. 5f, g). However, as observed for Dnmt1 and Dnmt3b, the developing joints showed highest expression levels for this gene, and the differentiating cartilage the lowest (Fig. 5g). Time course q-PCR expression analysis in samples of interdigital tissue (Fig. 5e, h, and k), revealed high expression levels of the three Dnmts up to id 6.5. From this stage, the expression of the three Dnmt´s underwent a progressive downregulation according to distinct temporal patterns (Fig. 5e, h, and k). By id 7.5, all Dnmts appeared downregulated below fifty percent of their starting levels, but DNMT proteins were still detected by western blots (Fig. 6a–c; Supplementary Fig. 3) and confirmed by immunocytochemistry (Fig. 6d–l). Comparative q-PCR analysis between digit rays and interdigit regions confirmed the predominant expression of Dnmt1 and Dnmt3b in the interdigits and, at a lower level, of Dnmt3a (Fig. 5l). Double immunolabeling for DNMTs and 5mC showed a widespread distribution of all DNMTs through the nucleoplasm (Fig. 6d, g, j). In addition, DNMT3B, often showed overlapping expression with 5mC foci (Fig. 6j–l). This expression pattern was conserved even at advanced stages of interdigit remodeling (id 7.5).

**Methylation in cell death and chondrogenic differentiation**. We selected DNMT3B, as the de novo methyltransferase with the highest expression level in the interdigit to explore its influence on digit skeletal progenitors. We analyzed the cellular outcome by using high-density micromass cultures, as a well-established model to study mesenchymal skeletal precursors differentiation. Dissociated progenitors transfected with vectors containing the coding region of the chicken Dnmt3b gene showed a significant increase in global methylation after two days of culture (Fig. 7a). Cell death at this time of culture increased almost three-fold in transfected progenitors (Fig. 7b–d), and chondrogenesis was

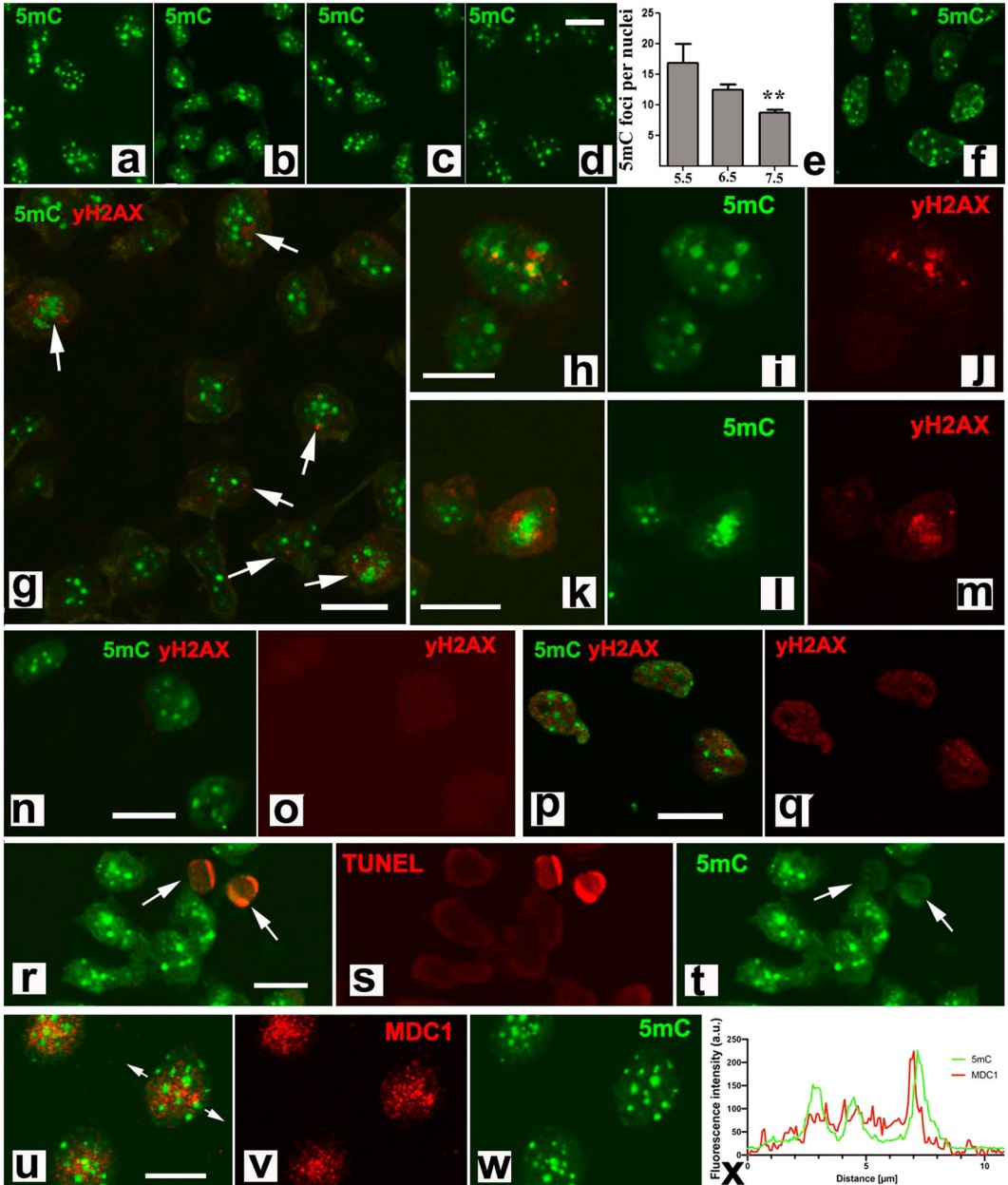

**Fig. 4 DNA methylation and DNA damage. a, b** representative illustrations of the conserved pattern of nuclear 5mC immunolabeling in dissociated interdigital mesodermal progenitors at id 5.5 (**a**), 6 (**b**), 7 (**c**), and 7.5 (**d**). **e** quantification of the number of 5mC foci per cell at id 5.5, 6.5, and 7.5 (n = 4 biologically independent experiments). **f** distribution of 5mC in dissociated mesodermal progenitors of the digit tip at id 6.5. **g** double immunolabeling with anti-5mC (green) and anti-γH2AX (red) in id 6.5 interdigital dissociated cells to show the preferential distribution of γH2AX marks in the contour of 5mC foci. Arrows indicate cells showing γH2AX-positive foci. **h–j** merged (**h**) and channel separated (**i, j**) detailed views of the nucleus of an interdigital progenitor at id 7.5, showing the overlapping distribution of γH2AX (red) and the 5mC foci (green). **k–m** double immunolabeling with anti-5mC (green) and anti-γH2AX (red) showing the association with, and increased distribution of, γH2AX in the 5mC aggregated foci. **n, o** 5.5 id control skeletal progenitors double immunolabeled with 5mC (green) and γH2AX (red). **n** merged imagen; **o** dissociated red channel. **p, q** 5.5 id skeletal progenitors 12 h after 2 Gy irradiation immunolabeled with anti-5mC (green) and γH2AX (red). **p** merged image; **q**, dissociated red channel. Note the absence of γH2AX foci in controls (**o**) and its widespread distribution in irradiated progenitors (**q**). **r–t** merged (**r**) and channel dissociated (**s, t**) images of interdigital progenitors at id 6.5 double labeled with anti-5mC (green) and TUNEL (red). Note the loss of 5mC foci in the interdigital cells TUNEL-positive (arrows). **u–w** double immunolabeling of dissociated interdigital cells at id 6.5 (**u**), showing the distribution of the DNA damage response factor MDC1 (red labeling, **v**) around the foci positive for 5mC (green labeling, **w**). **x** is a pixel intensity plot of the region indicated by white arrows in **u**, to confirm the distribution of MDC1 (red) in the periphery of the 5mC foci (green). Note that in all images of this panel staining is restricted to the nuclei and the cytoplasm is not appreciated. Bars = 6 μm.

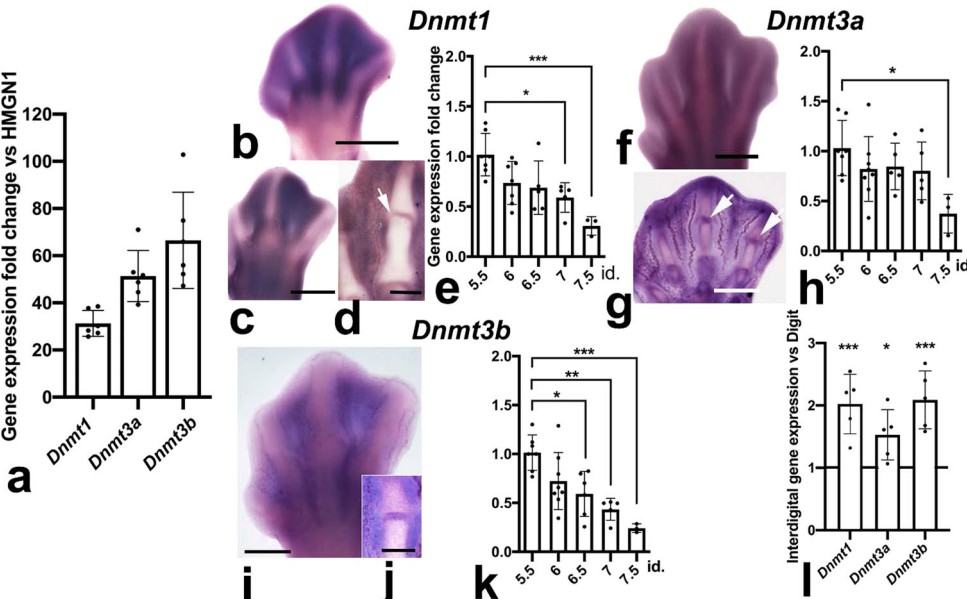

**Fig. 5 Expression analysis of *Dnmt* genes by in situ hybridization and q-PCR. a** q-PCR comparative analysis of the interdigital expression levels of *Dnmts* related to that of *HMGN1* (*n* = 6 biologically independent samples). **b**, **c** whole mount in situ hybridizations for *Dnmt1* of autopods at id 6.5 (**b**), and 7.5 (**c**). **d** longitudinal section of the autopod at id 7.5 illustrating the presence of *Dnmt1* expression domains in the developing joints (arrow). **e** q-PCR sequential changes in the level of interdigital expression of *Dnmt1* (*n* = 3–8 biologically independent samples). **f** whole mount in situ hybridizations for *Dnmt3a* at id 7.5. Note that interdigital expression is less intense than the others *Dnmt* genes. **g** expression of *Dnmt3a* in a longitudinal section of an autopod at id 6.5. Note the widespread distribution of transcripts with intensified domains in the developing joints (arrows). **h** q-PCR sequential changes in the level of interdigital expression of *Dnmt3a* (*n* = 3–8 biologically independent samples). **i**, **j** in situ hybridizations for *Dnmt3b* in whole mount (**i**) and in a sectioned digit (**j**) at id 7 and 7.5, respectively. Note the interdigital domains in **i**, and the intense expression in the joint interface and peridigital mesenchyme in **j**. **k** q-PCR sequential changes in the level of interdigital expression of *Dnmt3b* (*n* = 3–8 biologically independent samples). Note that, for comparative purposes, in the q-PCR plots (**e**, **h**, and **k**) the expression level of each gene at id 5.5 was arbitrarily considered 100%. **l** comparative q-PCR analysis of the level of expression of *Dnmt* genes between the third interdigit and digit 3 at id 6.5 (*n* = 5 biologically independent samples). For comparative purposes the expression levels in the digit rays were considered 1, and is represented by the horizontal black line. Note the predominant expression of *Dnmts* in the interdigital tissue and interphalangeal joint regions. Bar = 200 μm. ***$p < 0.001$; **$p < 0.01$; *$p < 0.05$.

reduced by three-fold (Fig. 7e, g, h). In a complementary fashion, DNA methylation was reduced when progenitors were transfected with a short hairpin RNAi against *Dnmt3b* (Fig. 7a), further, cell death was moderately, but significantly, reduced (Fig. 7d), and chondrogenesis increased by almost two-fold after 5 days of culture (Fig. 7f, i).

To investigate whether the biological effect of *Dnmt3b* gene silencing was related to methylation catalysis, we examined the effect of chemical inhibition of DNA methyltransferase activity with 5-azacytidine[21]. Indeed, consistent with the proposed influence of hypomethylation on the activation of prochondrogenic genes (see review by ref. [22]), the addition of 5-azacytidine to micromass cultures promoted chondrogenesis at levels similar to what was observed after *Dnmt3b* gene silencing (Fig. 7j–l). The intensity of cell death decreased after this treatment but without reaching statistically significant levels. This limited inhibitory effect on cell death might reflect the specificity of 5-azacytidine for DNMT1 rather than the de novo DNA methyltransferases 3A and 3B[23]. Alternatively, these results might suggest that the primary effect of DNMT′s in the micromass culture assay was associated with the regulation of chondrogenic differentiation.

**Regulation of key differentiation genes by DNMT3B.** To gain molecular insights into the mechanisms accounting for the effects of DNMT3B, we analyzed transcriptional changes in the molecular cascades associated with cell death, cell senescence, and skeletal tissue differentiation. We selected *p21* as a marker of senescence[24] and *Bak1*[25] as a marker of apoptotic activity. For cell

differentiation, we selected *Sox9* and *Scleraxis* due to their competing functions in skeletogenesis, promoting chondrogenesis and fibrogenesis, respectively[26,27]. As shown in Fig. 8a, overexpression of *Dnmt3b* was accompanied by overexpression of *p21* (fourfold), *Bak1* (more than two-fold) and *Scleraxis* (more than twofold), while *Sox9* was downregulated. Loss-of-function experiments caused mirror gene regulation, but at more moderated levels (Fig. 8b). These reduced regulatory effects could be explained because *Dnmt3b* gene silencing in loss-of-function experiments was also relatively moderate (0.5 x).

The close relationship between methylation and transcriptional activity prompted us to analyze by MSRE-q-PCR, differences in CpG methylation at the promoters of the selected genes. As shown in Fig. 8c–f, these genes exhibited methylation differences across their promoters. The promoter of *Sox9* was hypomethylated after *Dnmt3b* gene silencing, and methylation increased in gain-of-function experiments (Fig. 8f). This finding suggests that *Sox9* hypomethylation is characteristic of zones of chondrogenesis. Consistent with this interpretation, the promoter of *Sox9* also appeared hypomethylated in samples obtained from the chondrogenic tip of the growing digits of control embryos in comparison with interdigital tissue samples of the same stage (Fig. 8g). *Bak1*, *Scleraxis*, and *p21* showed promoter hypomethylation in the gain-of-function experiments that correlated with the increased gene expression level (Fig. 8c–e). In a complementary fashion, methylation across the promoter of these genes increased in loss-of-function experiments, which is consistent with their downregulation at the transcriptional level.

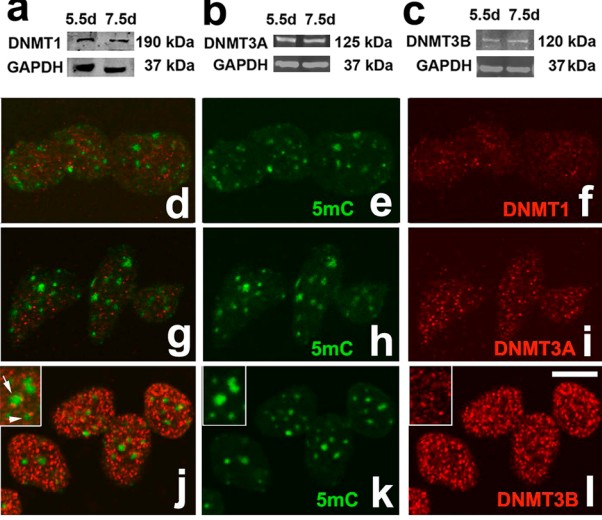

**Fig. 6 Expression of DNMT proteins in the interdigital mesoderm.**
**a**–**c** representative western blottings showing DNMT1 (**a**), DNMT3A (**b**), and DNMT3B (**c**) proteins in the interdigital tissue at id 5.5 and 7.5, the stages preceding and at the peak of degeneration, respectively. **d**–**l** Immunolabeling of dissociated interdigital progenitors at id 6.5, double labeled with anti-5mC (green in **d**, **e**, **g**, **h**, and **j**, **k**) and anti-DNMT1 (**d**, **f**; red); anti-DNMT3A (**g**, **i**; red) and DNMT3B (**j**, **l**; red). **d**, **g**, and **j** are merged images of **e**, **h**, and **k** (green labeling with anti-5mC) and **f**, **i**, and **l** (red labeling with each anti-DNMT' antibodies). Note the close correlation between the labeling intensity and the transcriptional levels illustrated in Fig. 5b–k. Insets in **j**–**l** illustrate in detail the association of DNMT3B with the 5mC foci. Bar = 5 μm.

## Discussion

Classical embryological studies performed during the second half of the last century identified cell death as a crucial developmental mechanism in metazoa (see, for example ref. [28]). The formation of free digits in the developing limb of vertebrates involves the elimination of the mesodermal progenitors located between the developing digit rays and is considered a paradigm of the so-called "programmed", "embryonic", or "morphogenetic" cell death because the level of cell death is correlated with the dual, free vs webbed, digit morphology in amniotes[6]. Initial genetic studies performed in the *C. elegans*, that were next extended to insects and vertebrates, discovered a conserved molecular machinery accounting for the elimination of the dying cells in developing systems (see ref. [29]). The cysteine-aspartic proteases (caspases), activated via permeabilization of the mitochondrial outer membrane (intrinsic death pathway), or, by specific cell surface receptors (extrinsic death pathway), were considered central players of the embryonic dying processes. Furthermore, the term of apoptosis was established to define a type of cell death subjected to genetic control and accounting for most embryonic dying physiological processes[30]. However, accumulating evidence has revealed the existence of multiple degenerative pathways, including lysosomal permeabilization and autophagy, cooperative phagocytosis, cell membrane disruption, or, even, cell senescence, which also participate in the elimination of cells and tissues in embryonic systems[1,31,32]. Most importantly, it was observed that in the regressing interdigits, these degenerative pathways participate in a redundant fashion[1,33]. These findings suggest that there are regulatory events upstream of the degenerating cascades. Recent expression and functional analysis of the major epigenetic regulators UHRF1 and 2[13] suggested the contribution of epigenetic mechanisms controlling interdigital cell death. The present

study uncovers the involvement of the epigenetic status of interdigital cells in the remodeling process via regulation by DNA-methyl transferases.

Our findings show that, at difference of the growing digit rays, the interdigital cells are very vulnerable to DNA damage by exposure to X-irradiation. Similar results were found by local treatments with BMP-beads that are considered the triggering signal of interdigital cell death. Studies on cancer cells have proposed global hypomethylation as a major factor responsible for sensitivity to DNA damage and subsequent death, and for the resistance to genotoxic therapies[34,35]. Here, we observed that the differential sensitivity to irradiation of interdigital and digit progenitors occurs with similar levels of global methylation, suggesting that the irradiation sensitivity of skeletal progenitors might be mediated via methylation/demethylation of selected CpG islands in target genes[36].

We observed here that the pattern of DNA methylation in the interdigit is associated with specific expression patterns of DNA-methyl transferases. According to our expression and function studies, mainly of DNMT3B, DNMT's function as positive regulators of cell death and negative regulators of chondrogenesis in the skeletal progenitors of the autopod. The treatments with the hypomethylating agents, such as 5-azacytidine, replicate the prochondrogenic effects of *Dnmt3b* gene silencing, indicating that the anti-chondrogenic influence of DNMTs is linked with catalysis of methylation. Whether the proapoptotic influence of DNMTs is associated with hypermethylation as reported in other systems[37,38], or if cell death is secondary to inhibition of chondrogenic differentiation requires further analysis. Previous studies linking methylation by DNMTs and cell death were controversial. Inhibition of DNA methyltransferases induces apoptosis[39,40] and senescence[41] in cancer tissues, while it inhibits cell death in other contexts[42]. In embryonic systems, retinal apoptosis has been associated with hypermethylation[43] and, in early Xenopus embryos overexpression of *Dnmt3a* and *Dnmt3b* induced apoptosis[44]. However, the high lethality of hypomethylated embryos supports a proapoptotic influence in embryonic progenitors[45], and hypomethylation induced by *Dnmt1* inactivation caused apoptosis in the embryonic lens placode[46,47] and in hepatic tissue of zebrafish embryos[48]. Together, these findings suggest that the involvement of DNMTs in embryonic cell death may be achieved by different mechanisms that are context dependent.

Our functional analysis revealed that the de novo methylation resulting from overexpression of *Dnmt3b* is accompanied by transcriptional regulation of master genes of cell differentiation (*Sox9* and *Scleraxis*), cell death (*Bak1*), and cell senescence (*p21*). Transcriptional downregulation of *Sox9* was associated with hypermethylation of its promoter. However, the observed upregulations of *Bak1* and *p21*, and *Scleraxis* were associated with hypomethylation of the CpG islands across their promoters. Studies conducted mainly on different cancer cell types have established a role for DNMTs acting in combination with multiple cofactors, in the repression of specific genes[49]. Our results, therefore, indicates that the positive regulation of the genes mentioned above might be explained by silencing of, still uncharacterized, transcriptional repressors of the above-mentioned genes. Together, these findings suggest that the initial methylation of target genes by DNMT3B triggers a cascade of transcriptional events of importance for determining the fate of the skeletal progenitors. Our interpretation does not discard additional epigenetic regulatory mechanisms. Previous studies[50], have shown that growth factors, such as Wnt and FGFs, can regulate *Sox9* methylation, via the *Dnmt3a* gene, which is crucial for establishing the dual fate of either survival or death, of limb skeletal progenitors. Furthermore, PRMT5, an arginine methyl transferase, has also been shown to sustain the undifferentiated

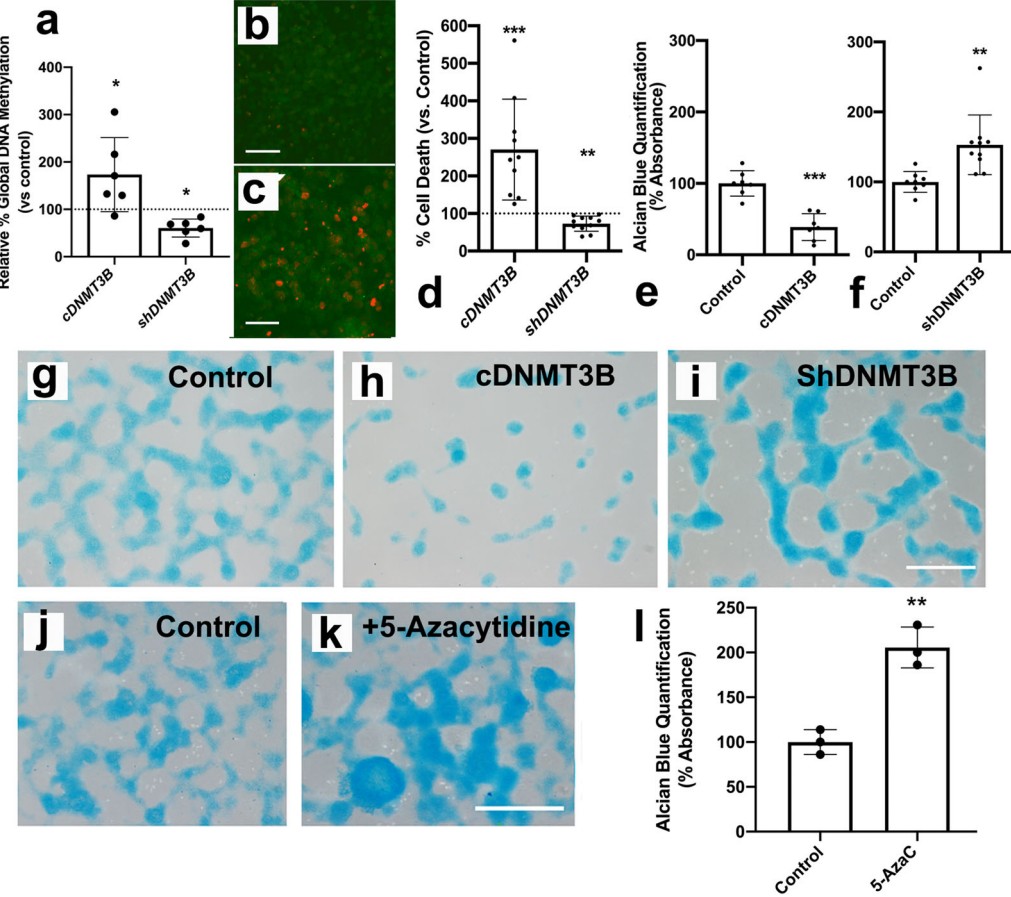

**Fig. 7 Gain- and loss-of-function analysis of *Dnmt3b* gene in limb skeletal progenitors cultured in high-density conditions. a** changes in global methylation evaluated by ELISA in interdigital progenitors overexpressing (*cDnmt3b*) or silencing (*shDnmt3b*) *Dnmt3b* gene (*n* = 6 biologically independent samples). **b** and **c** are representative confocal images showing the distribution of TUNEL-positive cells (red labeling) in control (**b**) and *Dnmt3b* overexpressing (**c**) progenitors at the end of the second day of culture. **d** shows the rate of cell death in cultures overexpressing *Dnmt3b* or subjected to gene silencing, evaluated by flow cytometry (*n* = 10 biologically independent samples). The dotted line represents the rate of cell death in the respective controls transfected with the expression vector only. **e**, **f** quantification of chondrogenesis by guanidine extraction of Alcian blue staining in 5 days micromass cultures (*n* = 7-10 biologically independent experiments) overexpressing *Dnmt3b* (**e**) and after gene silencing (**f**). **g-i** chondrogenesis evaluated by Alcian blue staining, in micromass cultures from experiments shown in **e** (**h**) and **f** (**i**). **j-l** illustrate the same staining in five days micromass cultures treated with 20 μM of 5-azacytidine (**k** compare with control in **j**) and its quantification by guanidine extraction of Alcian blue staining (**l**) (*n* = 3 biologically independent experiments). Bars = 40 μm (**b**, **c**); 200 μm (**g-k**). ***$p < 0.001$; **$p < 0.01$; *$p < 0.05$.

state of limb skeletal progenitors, protecting them from the dying influence of BMP4[51].

Among the potential factors that methylation might induce in skeletal progenitors are active changes in nuclear chromatin domains[52]. It is now well established that the regulation of 5mC patterns in embryonic cells is of major importance in the control of cell differentiation (reviewed by ref. [53]). DNA methylation, along with post-translational modifications of histones, determines the accessibility of the chromatin to transcription factors. In this study, the predominant distribution of γH2AX and other DNA damage (DNAD) repairing factors in association with 5mC marks suggests that the pattern of DNA methylation might generate weak regions in the DNA chain prone to breaking when subjected to mild stress. In this regard, it is remarkable that irradiation, that is an unspecific, and potent, genotoxic agent, induces a wider distribution of γH2AX marks, that differs from its preferential distribution in the contour of 5mC marks during physiological degeneration. This finding is consistent with the organization of chromatin in dynamic domains of distinct functional significance[52]. Remarkably, SOX9 appears to be a key factor in the

establishment of functional chromatin domains[54], and it may play a central role in the control of cell death in the embryonic limbs[55,56]. Furthermore, the DNAD repairing activity detected here in the progenitors destined to die indicates that interdigital cells attempt to protect themselves from local harmful agents, such as oxidative stress[5,6]. Considering the above findings together, we propose that rather than a "cell suicide", interdigital cells represent a characteristic "sabotage" according to the terminology recently proposed by Douglas R. Green[57].

In summary, our study provides conceptual advances to our understanding of the regulation of embryonic programmed cell death based on epigenetic mechanisms. According to our findings, cells fated to die are progenitors that are able to develop specific cartilage templates[58]. The distribution of the dying process is sculpted by the absence of initial differentiation signals, in combination with the cessation of signals required to sustain the survival of progenitors. Considering these facts, it is tempting to speculate that such a dual process generates a cell population with epigenetic characteristics of undifferentiated cells that collapse when subjected to signals appropriate for progenitors that have initiated chondrogenic differentiation, as has been observed in

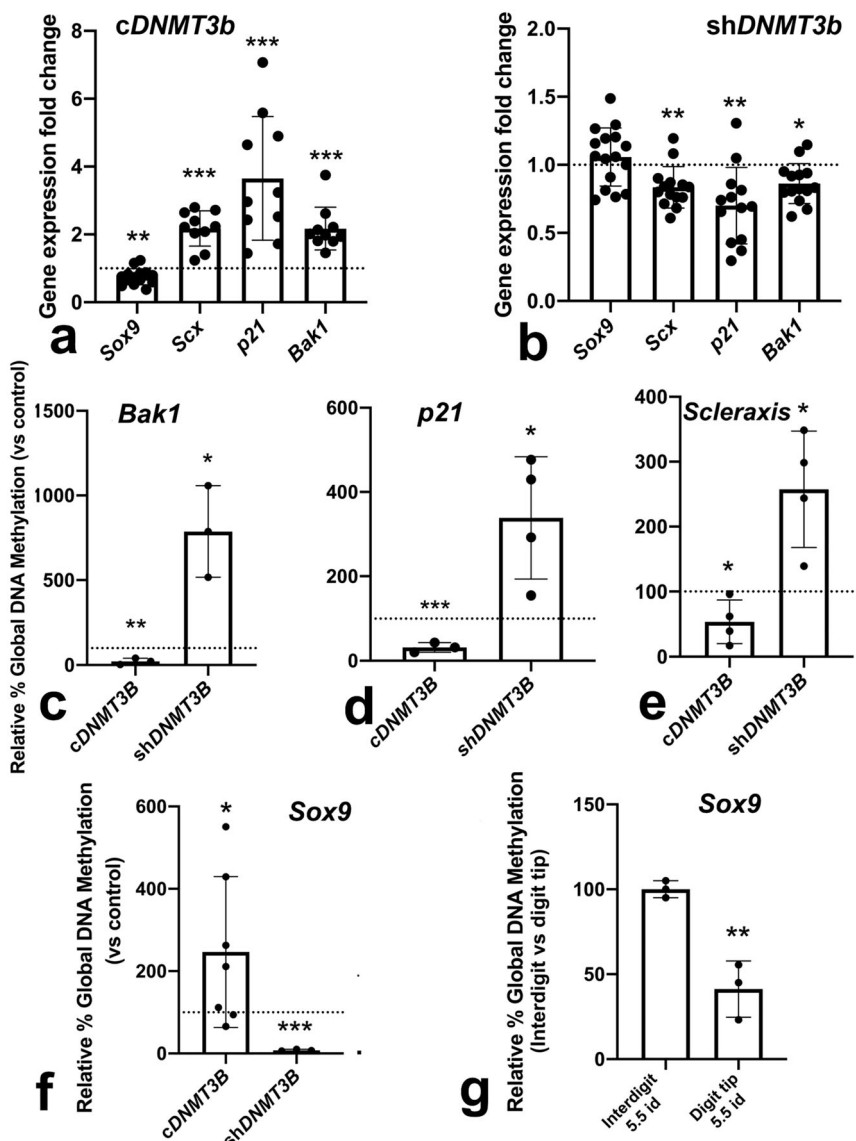

**Fig. 8 Charts showing gene regulation and promoter methylation after DNMT3B functional experiments in micromass cultures of skeletal progenitors.**
**a**, **b** transcriptional regulation of *Sox9*, *Scleraxis*, *p21*, and *Bak1* after *Dnmt3b* gene overexpression (**a** cDNMT3B, $n = 10$–14 biologically independent samples) and gene silencing (**b** shDNMT3B, $n = 13$–16 biologically independent samples). The level expression in the respective controls transfected with the expression vector only is represented by the dotted line. **c**–**f** methylation level of CpG islands in the promoter of *Bak1* (**c**), *p21* (**d**), *Scleraxis* (**e**), and *Sox9* (**f**) after *Dnmt3b* overexpression (cDNMT3B) or gene silencing (shDNMT3B), evaluated by MSRE-q-PCR ($n = 3$–8 biologically independent samples). **g** level of methylation of CpG islands of the *Sox9* promoter in samples of the interdigit (left bar) and digit tip (right bar) at id 5.5, showing the relative hypomethylation of the promoter of *Sox9* in the prechondrogenic cells of the digit tip in comparison with the adjacent interdigital progenitors ($n = 3$ biologically independent samples). ***$p < 0.001$; **$p < 0.01$; *$p < 0.05$.

mouse embryonic stem cells deficient in *Dnmt1*[45,46,59]. This implicates two important concepts: (1) the signals triggering the degenerative events, i.e., BMPs in the interdigits, are not true dead signals, but signals occupying advanced steps in the molecular differentiation cascade that would be unacceptable for naive progenitors; and, (2) the dying mechanism encoded by the often termed "death genes" is not specific for a particular death process; rather it represents the destructive machinery that cells are able to activate at a precise point in their development regardless of the primary dead-triggering signal.

## Methods

We employed Rhode Island chicken embryos (*Gallus gallus*) from day 4 to day 8.5 of incubation (id) equivalent to stages 23–34 HH.

**In situ hybridization.** In situ hybridization of PFA-fixed limb specimens was performed in whole mount or 100 μm vibratome sections. The samples were treated with 10 μg/ml of proteinase K for 20–30 min at 20 °C. Hybridization with digoxigenin-labeled antisense RNA probes was performed at 68 °C. Alkaline phosphatase-conjugated anti-digoxigenin antibody (dilution 1:2000) was used (Roche). Reactions were developed with BM Purple AP Substrate precipitation (Roche).

The probes for *Dnmt1*, *Dnmt3a*, and *Dnmt3b* were obtained by PCR from RNA extracted from chick limb buds at initial stages of digit formation. Specific primers for chick were the following: *Dnmt1*, Fwd: GAGGACTGCAACGTTCTGC Rev: TGCTGACGAACTTCTTGTCG; *Dnmt3a*, Fwd: GAGAGAGGCGGAGAAGAA GG Rev: TGTCAGTCTCGTCGTTCTCG; *Dnmt3b*, Fwd: ACGAAGATGGCTA CCAGTCC; Rev: TCTTGGTGATGTTCCTGACG

**Irradiation experiments.** Eggs were irradiated using an X-Ray generator system (Maxishot-d, Yxlon Int., USA) equipped with an X-Ray tube that works at 200 kV and 4.5 mA. After several exposures, we selected doses ranging from 1 to 4 Gy that

were not lethal for the embryos. Control and irradiated embryos were dissected free between 3 h and 3 days after irradiation and processed for further analysis.

**In vivo local applications of BMPs.** Eggs were windowed at id 5.5 and two heparin acrylic beads (Sigma) incubated in 5 μg/ml of recombinant BMP5 (R&D Systems) were implanted, at the same time, into the right autopod of the embryo. In each embryo, one bead was implanted into the third interdigit and the other in the tip of digit 3. For controls, the beads were incubated in PBS. The embryos were sacrificed 3 or 6 h after the treatment, and the autopod was dissected free, fixed in PFA and sectioned longitudinally with a vibratome. Sections were processed for immunofluorescent detection of DNA damage with anti-γH2AX antibody (see below) and counterstained with a rabbit polyclonal antibody against Sox9 (see below) to mark the nascent digit rays, and examined under the confocal microscope. The skeletal morphology and the pattern of cell death of the manipulated limbs was studied also at longer time periods in whole mount specimens after cartilage staining with Alcian blue or neutral red vital staining, respectively.

**Histological methodology.** Neutral red staining, alcian blue cartilage staining, and TUNEL assay, were performed as described previously[58].

Immunolabeling was performed in limb tissue samples fixed in 4% PFA. We employed dissociated cells obtained from squashed interdigital tissue fragments, or vibratome sections permeabilized with Triton X-100 in PBS. The following primary antibodies were employed (dilution 1:100): rabbit polyclonal anti-DNMT1 (NB100–26455; NovusBio.Co. USA) rabbit polyclonal anti-DNMT3A (ab2850; and ab188470; Abcam); rabbit polyclonal anti-DNMT3B (ab2851; Abcam); mouse monoclonal against 5mC (33D3 Eurogentec); rabbit polyclonal anti-MDC1 (ab41951; Abcam); mouse monoclonal anti-γH2AX (JBW301, Milipore-Upstate); and rabbit polyclonal anti-Sox9 (Merk-Millipore). Observation were made with a LSM51O laser confocal microscope (Zeiss).

For quantification experiments of 5mC foci or degenerating cells (γH2AX and TUNEL), at least 100 cells were counted in each individual experiment.

**Mesodermal cultures and in vitro treatments.** Dissociated undifferentiated mesoderm from chick leg autopods at 4.5 id were cultured at a density of $2.0 \times 10^7$ cells/ml to establish an organoid-like tridimensional growth system, that is termed "micromass culture". The chondrogenic outcome was studied under the microscope after Alcian blue staining (0.5% Alcian blue, at pH 1.0). Changes in chondrogenesis were further characterized quantifying optically the Alcian blue dye extracted in 6 M guanidine-HCl (pH 5.8). In each quantification experiment, measurements were obtained from pools of four stained micromass cultures. Chemical inhibition of DNA methyltransferase activity was performed by addition of 20 μM 5-azacytidin (A2385; Sigma–Aldrich) to the medium.

**Cell nucleofection and targeted gene silencing.** Genetic studies were performed by gain- and loss-of-function approaches.

For gain-of-function experiments, skeletal progenitors were electroporated with constructs of chicken Dnmt3b cloned into the pcDNA3.1 vector (GenScript). For loss-of-function experiments, skeletal progenitors were electroporated with a short hairpin RNAi against Dnmt3b (sh-Dnmt3b) cloned into the pcU6–1-shRNA (a generous gift from Dr Tim J. Doran). Electroporation was performed with a Multiporator System (Eppendorf) and cultured under high-density conditions as indicated above. Transfections with the respective empty plasmids were employed as controls. After 48 h of culture, the level of gene regulation was confirmed by q-PCR and/or western blot analysis. In overexpression experiments upregulation ranged between 13 and 60 folds (mean 25.20 ± 7). In gene silencing experiments downregulation ranged between 0.3 and 0.6 folds (mean 0.52 ± 0.04).

**Western blot analysis.** Tissue samples obtained from 30 interdigital regions of each stage were processed for western blot analysis ($n = 4$). Total proteins were extracted by lysis. After determining the protein concentration, 30 μg of each sample was loaded onto a 12.5% SDS polyacrylamide gel, electrophoresed and transferred to PVDF membranes. The membranes were incubated with primary antibodies (dilution 1:1000; see immunofluorescence section). Protein bands were detected with an Odyssey ™ Infrared-Imaging System (Li-Cor Biosciences), using anti-mouse IRDye800DX or anti-rabbit IRDye680DX as secondary antibodies (Rockland Immunochemicals, USA).

**Flow cytometry.** Cell death was evaluated by flow cytometry in dissociated cultured cells after the functional experiments. One million cells were used in each test. For propidium iodide (PI) staining, the cells were washed with PBS and fixed in 90% ethanol. The samples were incubated overnight at 4 °C with 0.1% sodium citrate, 0.01% Triton X-100 and 0.1 mg/ml PI. The cell suspension was subjected to flow cytometry analysis in a Cytoflex (Beckman Coulter) and analyzed with the Cytexpert software. The gating strategy has been performed by deselecting impurities (debris) employing an SSC-A / IP-PE dot plot and then selecting the apoptotic and living cells in a logarithmic scale IP-PE histogram (see Supplementary Fig. 4).

**Global methylation and promoter methylation.** Genomic DNA was extracted using NucleoSpin® Tissue (Macherey–Nagel). Gobal methylation was determined by the ELISA-based commercial kit Imprint® Methylated DNA Quantification (MDQ1; Sigma–Aldrich, St. Louis, MO, USA). One hundred and fifty nanograms of genomic DNA from our samples were incubated with capture and detection antibodies and their absorbance was measured at 450 nm. The amount of methylated DNA present in the samples is proportional to the absorbance measured. Quantification of global DNA methylation was performed calculating methylation levels relative to the methylated control DNA (50 ng/μl) using the formula: [(A450 av sample–A450 av blank)/ (A450 av methylated control DNA–A450 av blank)] × 100. Changes in the methylation status of Sox9, Scleraxis, Bak1, and p21 gene promoters were studied by methylation-sensitive restriction enzyme and quantitative polymerase chain reaction (MSRE-q-PCR). Primers flanking the region of interest (based on the presence of informative restriction sites) within the promoter of each gene were designed using Primer3Plus online software. The following primers were selected: Sox9: Fwd: CCATCACGCTCACA CTCTC; Rev: AGGTACCGCTGTAGGTGGTG; Scleraxis: Fwd: CTGTACCCCGA GATCAGCAT; Rev: GGTGTTGACGCTGTTGGTG; p21: Fwd: GCTATAAAG GGCGGAGTGC; Rev: CCATCACCCCCTCTTTCC; Bak1: Fwd: AGCTGCAGC CTTCCCAGA; Rev: CTCTAGAGGCGCCTTGCAC

Genomic DNA samples were digested with a CpG-methylation-sensitive restriction enzymes: TauI (3 U/μl; ER1651 ThermoFisher Scientific), or FauI (200U/μl; R0651S New England BioLabs); or with non-CpG-methylation-sensitive enzymes: SacI (10 U/μl; ER1132 ThermoFischer Scientific) or HphI (10U/μl; ER1102 ThermoFischer Scientific) for 2 h according to the manufacturer's suggested temperature. SYBRGreen-based q-PCR was carried out in triplicates with a total volume of 20 μl per tube containing 1 μl of genomic DNA (TauI-digested, SacI-digested, or undigested DNA), 0.4 μl of each primer, 10 μl of SYBR Select Master Mix (Life Technologies), and 8.2 μl of $H_2O$. Reactions were carried out in a StepOne Real Time System and analyzed by StepOne software v2.3 (Life Technologies). The relative percentage of methylated DNA was calculated according to the equation $2^{-\Delta\Delta Ct}$ employing as normalizers the Ct values of both SacI, or HphI digested and undigested samples.

**Real-time quantitative PCR (q-PCR) for gene expression analysis.** Total RNA was extracted using the NucleoSpin RNA kit (Macherey–Nagel). First-strand cDNA was synthesized using random hexamers and the High Capacity cDNA Reverse Transcription Kit (Life Technologies). The cDNA concentration was adjusted to 0.5 μg/μl. SYBRGreen-based q-PCR was performed as indicated above using the Mx3005P system (Stratagene). Primer-probe sets were designed employing the Primer3 software, and efficiency was verified in silico employing the "primer stats" and "primer product" tools (http://www.bioinformatics.org/sms2) and a nBLAST analysis (https://blast.ncbi.nlm.nih.gov/Blast.cgi). Specificity was checked by the presence of single peaks in the dissociation curves. In the case of chick Dnmt1, Dnmt3a, and Dnmt3b genes, expression efficiency was further validated analyzing samples from dorsal neural tube and neural crests regions microdissected from early embryos (id 1.5) were expression of Dnmts has been previously well documented[19,20]. Rpl13 was chosen as the normalizer in interdigital samples and Gapdh in cultures. Mean values for fold changes were calculated. Gene expression levels were evaluated relative to a calibrator according to the $2^{-\Delta\Delta Ct}$ equation. q-PCR chicken specific primers were: for Gapdh, Fwd: GGTGGCCATC AATGATCC and Rev: GTTCTCAGCCTTGACAGTGC; for Rp13, Fwd: AACTC AAGATGGCAACTCAGC and Rev: AAGGCCTTGAAGTTCTTCTCC; for Dnmt1, Fwd: CACAGCCAGCACAAGTTCC and Rev: TGAGCAGGTAGCGTTG AAGG; for Dnmt3a, Fwd: GCAGGATAGCCAAGTTCAGC and Rev: CACAGG ATGTCTTCCTTCTCG; for Dnmt3b, Fwd: CAAGAGGCTGAAGAGCAACC and Rev: CGCTGTTGTTCGTAACTTCG; for Bak1, Fwd: CTACGTCACCGAATTC ATGC and Rev: AACATTGTCCAGATCGAGTGC; for p21, Fwd: CGTAGACC ACGAGCAGATCC and Rev: CGTCTCGGTCTCGAAGTTG; for Sox9, Fwd: GAGGAAGTCGGTGAAGAACG and Rev: GATGCTGGAGGATGACTGC; for Scleraxis, Fwd: CACCAACAGCGTCAACACC and Rev: CGTCTCGATCTTGG ACAGC.

**Statistics and reproducibility.** All the measurements included at least three independent experiments performed under the same conditions. These replication attempts were successful. Statistical analyses were performed using GraphPad Prism 7 (GraphPad). The results are expressed as means ± standard deviation. The significance of differences among three or more groups was evaluated through one-way analysis of variance ANOVA with post-hoc Bonferroni test. Paired analyses were performed using Student's $t$-test. *$p < 0.05$, **$p < 0.01$, and ***$p < 0.001$ denoted statistical significance.

**Ethics statement.** All experiments were carried out under the standards of experimentation dictated by the European Community (2010/63/EU) in accordance with Spanish legislation (RD53/2013) and approved by the ethics committee of the local government through the authorization corresponding to our research project (PI-03–18).

**Reporting summary**. Further information on research design is available in the Nature Research Reporting Summary linked to this article.

## Data availability

The datasets analysed during the current study are available in the Figshare repository, at https://doi.org/10.6084/m9.figshare.12263843

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

## Acknowledgements

We thank Prof. Miguel Lafarga for helpful comments and advice. We thank Dr Jose E Gomez-Arozamena for helping us with the irradiation experiments. We are grateful to Montse Fernandez Calderon, Susana Dawalibi, and Sonia Perez Mantecon, for excellent technical assistance. This work was supported by a Grant (BFU2017–84046-P) from the Spanish Science and Innovation Ministry to JAM. C.S.F is recipient of a FPI grant (BES-2015–074267).

## Author contributions

C.S.F and C.I.L.D. performed the experiments and collected the results. C.S.F., C.I.L.D., J.M.H., and J.A.M. designed the experiments and interpreted the results. J.M.H. and J.A.M. wrote the manuscript.

## Competing interests

The authors declare no competing interests.
