## [Peer Review File · Communications Biology]

Reviewers' comments:

Reviewer #1 (Remarks to the Author):

In this study, Sanchez-Fernandez et al. analyze a possible link between DNA methylation and interdigital cell death. Interdigital cell death is acknowledged as one of the prime examples of developmental tissue sculpting by apoptosis. The authors first demonstrate that stress induced interdigital apoptosis is preceded by DNA damage. Following this, the authors aim to correlate DNA damage and apoptosis in interdigital cells with DNA methylation. They further show that all three DNA methyltransferases are expressed in interdigital tissue *in vivo*. Finally, chicken micromass cultures are used to functionally test a role for Dnmt3b in limb mesenchymal progenitor cells. This reveals, that Dnmt3b represses the chondrogenic program. This coincides with hypermethylation of the Sox9 promoter.

The study nicely demonstrates a specific vulnerability of interdigital cells to exogenous stress, leading to DNA damage. The authors hypothesize that this is conferred by the activity of DNA methyltransferases. After showing correlative evidence for DNA methyltransferase expression and 5'mC distribution with DNA damage, the story develops inherent inconsistencies. A possible role for 5'mC in vulnerability of DNA to stress signals is not substantiated by the following functional analysis of Dnmt3b, where the authors rather follow up a story of altered interdigital cell differentiation. Functional experiments of Dnmt3b overexpression and knockdown do not link Dnmt3b levels / 5'mC to DNA damage and apoptosis. Neither γ H2AX nor apoptosis was tested in this assay, which I do not understand. Thus this manuscript touches on different events in the course of interdigit regression, however without functionally linking them. This study clearly has potential, however several claims made in the abstract in my view are not backed substantially by experimental evidence. As major concerns, this reviewer regards a) the association of 5'mC with DNA damage (comments 4, 6); b) the clear lack of functional association of DNA methylation and DNA damage / apoptosis (comments 13, 15); c) induction of DNA damage and cell death by irradiation and BMP are not followed up when analyzing the role of DNA methylation (comments 4, 18). In the following, points that need to be addressed are outlined.

Specific comments:

1. P5: „...were the predominant degenerative feature of the autopodial tissue up to 6 hr after irradiation (Fig. 1C). TUNEL positive cells appeared after 6 hr of irradiation...”
The 6 hr time point is not shown in the figure. It would be very informative to show a time series of these events.
2. Fig. 1A, C, D: non-irradiated controls should be shown.
3. Comparing Figs 2 and 3A-D, it appears to me that the size as well as the number of 5'mC foci in dissociated interdigital cells is decreasing over time, which would correlate to the global 5'mC analysis. This should be quantified (dot size, numbers; or e.g. pixel intensities). Also, Fig 3 A-D should be sided by images (at least as supplementary data) showing the normal course of γ H2AX and TUNEL occurrence in chick interdigits at the stages chosen for analysis. In previous figures, γ H2AX is only shown after irradiation and BMP bead application.
4. Fig. 3: the association of 5'mC and γ H2AX is shown in a single image. This has to be supported by statistical evaluation over large cell numbers from several biological replicates to be credible. Furthermore, the experiment should be repeated with interdigital cells after irradiation. Fig. 3G: I do not see the association of MDC1 with the 5'mC foci. At least, single canal images should be shown so that overlap can be appreciated.
5. Fig. 3 H, I: the stage the cells were taken from is not indicated in text of figure legend.

6. Figure 3 lacks a control tissue; the same analysis should be performed on dissociated digit cells.
7. Fig. 4A: in my view, the method the authors used is not an absolute quantification. Have the primer /amplification efficiencies been tested and compared? To appreciate whether interdigital tissue shows high expression of Dnmts, other tissues should be tested for comparison.
- 8: Fig. 4: I find it very hard to appreciate the exact expression patterns of the DNMT genes in the whole mount ISH images provided, especially Dnmt3b. This should be complemented by section ISH. Expression levels of Dnmt genes in interdigit tissue should be compared to digit tissue. The authors nicely show a functional role for Dnmt3b in repressing chondrogenesis and Sox9 expression. However the in vivo evidence for such a function is weak. At the least, lower expression of Dnmts in cartilage should be convincingly shown (section ISH, qPCR). See also comment 18.
9. Fig. 5A-C: it is not indicated how often this analysis was repeated.
10. In general, numbers of biological replicates used should be indicated in the figure legends.
11. Overexpression and knockdown of Dnmt3b: overexpression levels and knockdown efficacy have not been assessed. This should be checked by RT-qPCR.
12. Fig. 5F: I do not see the overlap between Dnmt3b and 5mC foci. Also a quantification of this is lacking.
13. Fig. 6B: cell death levels were analyzed by propidium iodine labelling followed by flow cytometry. This is an inaccurate method to assess cell death and should be backed up by alternative methods, preferentially methods that assess apoptosis, since this is the process in question.
14. Fig. 5C: increased chondrogenesis was seen after Dnmt3b knockdown; this would predict that decreased DNA methylation favors chondrogenesis. This is conflicting with equal global methylation levels in interdigits and digits (Fig. 2B). Can the authors please comment?
15. For me a major shortcoming in this manuscript is that DNA methylation is not functionally linked to DNA damage. At least γ H2AX as marker for DNA damage should be tested in this scenario. The authors note (p12) "In this study, the specific distribution of γ H2AX and other DNA damage (DNAD) repairing factors in association with 5mC marks suggest that the pattern of DNA methylation might generate weak regions in the DNA chain prone to breaking when subjected to stress.". This could and should be tested.
16. Dnmt3b overexpression is followed by hypermethylation of the Sox9 promoter, Sox9 downregulation is in line with the prevalent suppressive role of methylation; however, promoters of Bak etc were hypomethylated after Dnmt3b overexpression. Can the authors please discuss how this may be caused?
17. The authors state (p12) "This dual process generates a surplus cell population with epigenetic characteristics of undifferentiated cells...". I do not see this statement backed by the data.
18. Why is the micromass system with Dnmt3b overexpression and knockdown not used to test whether these manipulations confer mesenchymal cells with differential ability to respond to BMP stimulation? Does Dnmt overexpression favor cell death upon Bmp treatment, does knockdown favor chondrogenesis? This would link this dataset with Fig. 1.

Minor comments:

Fig. 1F "SOX9" is annotated in the image, but not γ H2AX

P29 l801 "merge imagen" should read "merged images"

The authors are inconsequent in gene / protein nomenclature (sometimes use of capitals for genes, sometimes not / sometimes use of italic for genes, sometimes not) in the body text as well as in annotations in figures (e.g. Fig. 4 shows qPCR >> nucleic acid >> should be italic; in 4A gene names are all caps, in B-C not).

While in Fig.2 and the results section, the authors correctly refer to "global DNA methylation" assessed by the assay they chose; in the abstract, however, the authors claim analysis of "genome wide" DNA methylation. This is an important difference, since the latter to me would imply the utilization of true, genome wide technologies as bisulfite sequencing.

General remark: I would strongly encourage indicating the developmental stages on the images in Figures 1, 3, 4, 5. This makes it easier to comprehend the figure at a glance.

Reviewer #2 (Remarks to the Author):

This is a nicely performed study in which the authors compare the effects of ionizing radiation and implantation of a BMP-laden bead into either the digit or interdigit region of the developing limb bud. The authors document that cells in the interdigit regions (but not the chondrogenic digit regions) respond to both these perturbations with high levels of DNA damage and cell death. In addition, the authors document that DNMT1, DNMT3A, and DNMT3B are all more highly expressed in the interdigital mesenchyme in comparison to the forming digits. While over-expression of DNMT3B in micromass cultures of limb bud mesenchymal cells increased global DNA methylation, and simultaneously increased cell death and blocked chondrogenic differentiation; over-expression of an shDNMT3B had the opposite effect. The authors make the argument that relative levels of DNA methylation distinguish whether limb bud mesenchymal cells will either undergo chondrogenesis or cell death. I think that this work is in principal suitable for publication in Nature Communications Biology. However, prior to publication I recommend that the authors determine whether altering global DNA methylation levels by over-expression of DNMT3B is directly inducing genes that lead to DNA damage/cell death or indirectly do so by specifically down-regulating the expression of Sox9. The authors could address this issue by determining whether over-expression of a Sox9 expression construct (viral or electroporated) can block the ability of ectopic DNMT3B to induce cell death and block chondrogenesis.

Reviewer #3 (Remarks to the Author):

"The methylation status of the embryonic limb skeletal progenitors determines their commitment to either interdigital cell death or digit cartilage differentiation", Sanchez-Fernandez et al submission to Communications Biology.

In this manuscript, Sanchez-Fernandez et al explore the role of epigenetics in the decision of skeletal progenitors in the embryonic autopod to undergo either programmed cell death or chondrogenesis. They previously reported that DNA breakage is one of the earliest signs of interdigital cell death and now present data including irradiation experiments, analysis of methylation status, and DNA methyltransferase gain- and loss-of-function studies to conclude that interdigital cell death is under epigenetic control.

Comments:

- Beyond the Methods section, the authors have made almost no reference to the species being

used. The chick model should be explicitly stated in both the abstract and the introduction. Species used in experiments referred to as background should also be noted if mouse, as it remains unknown if there are species specific differences in regulation of interdigital cell death.

- Authors should write out “days of incubation” at first use of i.d. abbreviation in main text, not just the methods section. Methods should have more info for the correlation of id and HH autopod stages, for those who use HH staging.
- Figure 1: the experimental design of this experiment is not clear. The authors mention sublethal doses are 1 to 4 Gy, but then sometimes use 3-4Gy and other times 1-2Gy. Why are the authors using a range of doses instead of one specific dose? Figures 1A,C,D+E should have control, non-irradiated autopod data for comparison, and ideally IHC for a range of irradiation doses.
- SupFigure 1: A “no irradiation” control should be included as well as number and qualification of phenotypes observed. How does this data fit in with existing chick limb irradiation studies (ex. Galloway et al Nature 2009)?
- Figure 2: It is unclear whether this experiment includes irradiation or not? This reviewer presumes the analysis is in untreated limbs, but the first sentence of the section alludes to irradiation. Panels 2A+B should have titles, and data global methylation data should also be presented for digit sample at all stages to allow for direct comparison of digit to interdigit over the whole time course.
- The authors are making a case that the interdigital cells are sensitized to undergo cell death by these epigenetic marks while the digit progenitors are not. With this in mind, the IHC in Figure 3 should also include digit cells as a control.
- The data in Figure 6 is quite compelling but it begs the question of whether the same effect would be found in vivo—have the authors tried expressing these constructs via in ovo electroporation or local in vivo treatment with 5-aza? Separately, it is unclear which vector control is being used as the dotted lines in Fig 6A+B (and same comment in Fig 7).

“The methylation status of the embryonic limb skeletal progenitors determines their commitment to either interdigital cell death or digit cartilage differentiation”, Sanchez-Fernandez et al submission to *Communications Biology*.

In this manuscript, Sanchez-Fernandez et al explore the role of epigenetics in the decision of skeletal progenitors in the embryonic autopod to undergo either programmed cell death or chondrogenesis. They previously reported that DNA breakage is one of the earliest signs of interdigital cell death and now present data including irradiation experiments, analysis of methylation status, and DNA methyltransferase gain- and loss-of-function studies to conclude that interdigital cell death is under epigenetic control.

Comments:

- Beyond the Methods section, the authors have made almost no reference to the species being used. The chick model should be explicitly stated in both the abstract and the introduction. Species used in experiments referred to as background should also be noted if mouse, as it remains unknown if there are species specific differences in regulation of interdigital cell death.
- Authors should write out “days of incubation” at first use of i.d. abbreviation in main text, not just the methods section. Methods should have more info for the correlation of id and HH autopod stages, for those who use HH staging.
- Figure 1: the experimental design of this experiment is not clear. The authors mention sublethal doses are 1 to 4 Gy, but then sometimes use 3-4Gy and other times 1-2Gy. Why are the authors using a range of doses instead of one specific dose? Figures 1A,C,D+E should have control, non-irradiated autopod data for comparison, and ideally IHC for a range of irradiation doses.
- SupFigure 1: A “no irradiation” control should be included as well as number and qualification of phenotypes observed. How does this data fit in with existing chick limb irradiation studies (ex. Galloway et al *Nature* 2009)?
- Figure 2: It is unclear whether this experiment includes irradiation or not? This reviewer presumes the analysis is in untreated limbs, but the first sentence of the section alludes to irradiation. Panels 2A+B should have titles, and data global methylation data should also be presented for digit sample at all stages to allow for direct comparison of digit to interdigit over the whole time course.
- The authors are making a case that the interdigital cells are sensitized to undergo cell death by these epigenetic marks while the digit progenitors are not. With this in mind, the IHC in Figure 3 should also include digit cells as a control.
- The data in Figure 6 is quite compelling but it begs the question of whether the same effect would be found in vivo—have the authors tried expressing these constructs via in ovo electroporation or local in vivo treatment with 5-aza? Separately, it is unclear which vector control is being used as the dotted lines in Fig 6A+B (and same comment in Fig 7).

REFEREE RESPONSE'S LETTER.

Please, find bellow our response to the referees and editorial comments on our manuscript “**The methylation status of the embryonic limb skeletal progenitors determines their commitment to either interdigital cell death or digit cartilage differentiation (COMMSBIO-19-1866-T)**”. We want to thank the referees and editor positive comments and suggestions. We have now included an important amount of new data taking into account most of them. We think that the new manuscript has improved considerably after the revision process.

Kind regards,

Juan A Montero

Reviewer 1. Specific comments:

1. P5: „...were the predominant degenerative feature of the autopodial tissue up to 6 hr after irradiation (Fig. 1C). TUNEL positive cells appeared after 6 hr of irradiation...” The 6 hr time point is not shown in the figure. It would be very informative to show a time series of these events.

We have addressed in detail this question. By using ImageJ software, we now provide a quantification of the different types of cells (γ H2AX-positive; TUNEL-positive; and, double γ H2AX/TUNEL positive cells) after irradiation with 2Gy. These data are reflected in new Figures 1D, and 1E,1F, and, 1G.

2. Fig. 1A, C, D: non-irradiated controls should be shown.

Taking into account the referee's suggestion our new figure 1 A shows an illustrative control (no irradiated) lacking degeneration.

3. Comparing Figs 2 and 3A-D, it appears to me that the size as well as the number of 5'mC foci in dissociated interdigital cells is decreasing over time, which would correlate to the global 5'mC analysis. This should be quantified (dot size, numbers; or e.g. pixel intensities). Also, Fig 3 A-D should be sided by images (at least as supplementary data) showing the normal course of γ H2AX and TUNEL occurrence in chick interdigits at the stages chosen for analysis. In previous figures, γ H2AX is only shown after irradiation and BMP bead application.

We have addressed these points in our revised version:

We agree with the referee in his/her appreciation and we have included new data. We added a new plot showing the pattern of global methylation in digit samples (new Figure 3B). In addition, we included a quantification of the number of 5mC foci per cell (new figure 4E) and also a new picture showing the pattern of 5mC in cells from the digit tip (new figure 4F). Finally, we also provide a new Supplementary Figure 2, illustrating the sequence of interdigital degeneration as suggested by the reviewer.

4. Fig. 3: the association of 5'mC and γ H2AX is shown in a single image. This has to be supported by statistical evaluation over large cell numbers from several biological replicates to be credible. Furthermore, the experiment should be repeated with interdigital cells after irradiation. Fig. 3G: I do not see the association of MDC1 with the 5'mC foci. At least, single canal images should be shown so that overlap can be appreciated.

We agree with the referee and we now try to make strongest case of our observation including new supportive data in this version of the manuscript.

We now provide a quantification of the association of 5mC and γ H2AX within the text. Additionally, we have substituted the former figure 3E for three distinct new images (new figures 4G, H and I) that we find convincing and representative. We also included a panel of figures at the end of this letter, to further reinforce the reviewer confidence in our claim.

We provide a new picture of 5mC/ γ H2AX labeling in control and irradiated cells (new figures 4J and K). The observations indicated that DNA breakage after irradiation is wider than in physiological degeneration. We discuss this aspect in the text.

Finally, we now include a more detailed illustration of the pattern of MDC1/5mC labeling, including single channel images and a pixel intensity quantification plot along the nucleus (new figures 4 M, M', M''; and 4N)

5. Fig. 3 H, I: the stage the cells were taken from is not indicated in text of figure legend.

We added that information (id 6.5).

6. Figure 3 lacks a control tissue; the same analysis should be performed on dissociated digit cells.

The new figure 4J is a control of cells of the digit tip lacking degenerative events. Furthermore, the new Figure 4G shows both degenerating and non-degenerating cells on the interdigital tissue.

7. Fig. 4A: in my view, the method the authors used is not an absolute quantification. Have the primer /amplification efficiencies been tested and compared? To appreciate whether interdigital tissue shows high expression of Dnmts, other tissues should be tested for comparison.

We now provide a detailed description of the q-PCR methodology and, as suggested by the reviewer, we have performed a similar quantification analysis in samples obtained from the dorsal neural tissue of early embryos, where the specific domains of *Dnmt3a* and *Dnmt3b* gene expression has been previously published (see references Hu et al., 2012, Genes Dev. 26: 2380; Hu et al., 2014, Proc Natl Acad Sci U S A. 111:17911). We mention in the results that Cts are almost identical to those detected in the mentioned early neural samples.

8: Fig. 4: I find it very hard to appreciate the exact expression patterns of the DNMT genes in the whole mount ISH images provided, especially Dnmt3b. This should be complemented by section ISH. Expression levels of Dnmt genes in interdigit tissue should be compared to digit tissue. The authors nicely show a functional role for Dnmt3b in repressing chondrogenesis and Sox9 expression. However, the in vivo evidence for such a function is weak. At the least, lower expression of Dnmts in cartilage should be convincingly shown (section ISH, qPCR). See also comment 18.

We fully addressed these questions. We have changed all the *in situ hybridization* images for new ones (new figures 5B,C,D,F,G,I,J). In addition, we provide a comparative Digit versus Interdigit analysis of *Dnmt* gene expression (new figure 5L). As the reviewer will note, we changed a bit the text reporting the expression of *Dnmt3a* to provide a more precise description.

9. Fig. 5A-C: it is not indicated how often this analysis was repeated.

The number of replicates is now included in the method section.

10. In general, numbers of biological replicates used should be indicated in the figure legends.

We tried to do that in the new version of our manuscript, but it is hard to mention a precise number of observations when morphological approaches are employed.

11. Overexpression and knockdown of *Dnmt3b*: overexpression levels and knockdown efficacy have not been assessed. This should be checked by RT-qPCR.

These quantifications were performed and the data are now included in the “methods” section.

12. Fig. 5F: I do not see the overlap between *Dnmt3b* and 5’mC foci. Also, a quantification of this is lacking.

We have included a detailed image of 5mC/DNMT3B labeling as an inset (new figure 6F). We think that considering the wider distribution of DNMTB dots, this imagen is enough representative.

13. Fig. 6B: cell death levels were analyzed by propidium iodine labelling followed by flow cytometry. This is an inaccurate method to assess cell death and should be backed up by alternative methods, preferentially methods that assess apoptosis, since this is the process in question.

We decided to maintain the data illustrated by Figure 7 (former Fig 6), but, following the reviewer suggestion, we now include a TUNEL confocal picture that support the increased cell death in the experiments of gain-of-function (new Fig. 7 B-B’). Propidium iodine/flow cytometry has been employed widely to quantify cell cycle changes and differences in apoptosis by many different researchers. We have experience with this technique, and the findings observed are fully consistent with other observations of the study.

14. Fig. 5C: increased chondrogenesis was seen after *Dnmt3b* knockdown; this would predict that decreased DNA methylation favors chondrogenesis. This is conflicting with equal global methylation levels in interdigits and digits (Fig. 2B). Can the authors please comment?

The reviewer is right. Our explanation is that, rather than differences in global methylation, changes in the pattern of methylation of the CpG islands of specific genes may explain the different cell behaviors (cell death versus cell differentiation). As shown in figure 8G of the revised manuscript, methylation of the promoter of *Sox9* is significantly lower in the digit tips, and it is hypermethylated and hypomethylated in *Dnmt3b* gain- or loss-of-fuction experiments respectively (Fig. 8F). It must be taken into account that SOX9 has an architectural function in the organization of the chromatin, binding to genomic regions containing H3K27ac but not H3K27me³ (Liu et al., 2018. Development 145: dev164459. doi: 10.1242/dev.164459) and interacting with the high mobility group N (HGMN) non-histone chromosomal proteins (Furusawa et al., 2006; Mol Cell Biol. 26:592).

15. For me a major shortcoming in this manuscript is that DNA methylation is not functionally linked to DNA damage. At least γ H2AX as marker for DNA damage should be tested in this scenario. The authors note (p12) “In this study, the specific distribution of γ H2AX and other DNA damage (DNAD) repairing factors in association with 5mC marks suggest that the pattern of DNA methylation might generate weak regions in the DNA chain prone to breaking when subjected to stress.”. This could and should be tested.

We think that the new quantification data performed with the ImageJ software support our claim: The association of 5mC foci with markers of DNA damage at the stages of overt degeneration appeared quite clear: *“The analysis of the distribution of a total of 576 γ H2AX foci from ten different samples of interdigital tissue at id 7.5, showed that 386 (67%) of the foci were located in or around the 5 mC foci and only 190 (33%), were far from them (more than 0.1 μ m). This difference is more remarkable considering that the 5mC foci occupied less than 10 % of the total surface of the nuclei.”*

16. Dnmt3b overexpression is followed by hypermethylation of the Sox9 promoter, Sox9 downregulation is in line with the prevalent suppressive role of methylation; however, promoters of Bak etc were hypomethylated after Dnmt3b overexpression. Can the authors please discuss how this may be caused?

We agree with the reviewer. We could not find reports in previous studies showing that overexpression of DNMTs resulted in gene up-regulation. Following the reviewer suggestion, we have discussed the results as follows: *“However, the observed upregulations of Bak1 and p21, and Scleraxis were associated with hypomethylation of the CpG islands across their promoters. Studies conducted mainly on different cancer cell types have established a role for DNMTs acting in combination with multiple co-factors, in the repression of specific genes. Our results, therefore, indicates that the positive regulation of the genes mentioned above might be explained by silencing of, still uncharacterized, transcriptional repressors of the above-mentioned genes”*.

17. The authors state (p12) “This dual process generates a surplus cell population with epigenetic characteristics of undifferentiated cells...”. I do not see this statement backed by the data.

Indeed, considering the comment of the reviewer, we have changed the text to mention that such interpretation is a tempting speculation.

18. Why is the micromass system with Dnmt3b overexpression and knockdown not used to test whether these manipulations confer mesenchymal cells with differential ability to respond to BMP stimulation? Does Dnmt overexpression favor cell death upon Bmp treatment, does knockdown favor chondrogenesis? This would link this dataset with Fig. 1.

We think that this was a good suggestion. We initiated the study and performed a considerable number of experiments but we had to interrupt the research when the Spanish Government closed the universities as part of the quarantine for the coronavirus. We are now out of the lab, and human resources and instruments (e.g. PCR machines) were transferred to the hospital to help with the diagnosis of the disease. We are afraid that we won't be able to restart our lab before some months has passed and everything get to normal. However, the results were quite clear so far, showing that in undifferentiated cells (cultures subjected to an initial 24 hr FGF treatment to maintain cells undifferentiated, following the protocol by Yokouchi et al., 1996, Development. 1996, 122:3725) BMPs potentiate the pro-apoptotic influence of *Dnmt3b* overexpression, but in conventional cultures, BMPs antagonized the death-promoting influence of *Dnmt3b* overexpression, inducing cartilage growth. However, we did not yet finish the required number of replicates and we will be unable to finish it within the next months. In addition, we think that this in vitro approach could be employed to explore other secreted factors

that we know are implicated in different aspects of the control of interdigital tissue remodeling (i.e, Wnt signaling, Retinoic acid signaling, Tgfbeta signaling). It is also appropriated to explore differences between the interdigits progenitors of the chick and the webbed foot of duck embryos. This will allow us to elaborate in the next future a comprehensive analysis of the interplay of secreted signals to characterize their impact, via epigenetic factors, in the evolutionary diversification of the digit morphology. We believe that everything could fit into a good study in the future, supporting and expanding the findings of the present manuscript. We have changed a bit the discussion, to note that cell differentiation may be the central aspect of the epigenetic regulation of interdigit remodeling.

Minor comments:

Fig. 1F “SOX9” is annotated in the image, but not γ H2AX

This has been solved in the new manuscript

P29 I801 “merge imagen” should read “merged images”

This has been solved in the new manuscript

The authors are inconsequent in gene / protein nomenclature (sometimes use of capitals for genes, sometimes not / sometimes use of italic for genes, sometimes not) in the body text as well as in annotations in figures (e.g. Fig. 4 shows qPCR >> nucleic acid >> should be italic; in 4A gene names are all caps, in B-C not).

We are sorry for that mistake that has now been corrected. We always tried to use the correct nomenclature.

While in Fig.2 and the results section, the authors correctly refer to “global DNA methylation” assessed by the assay they chose; in the abstract, however, the authors claim analysis of “genome wide” DNA methylation. This is an important difference, since the latter to me would imply the utilization of true, genome wide technologies as bisulfite sequencing.

The reviewer is right, we did the correction in our new version.

General remark: I would strongly encourage indicating the developmental stages on the images in Figures 1, 3, 4, 5. This makes it easier to comprehend the figure at a glance.

Thank you very much for this advice. We have now included the developmental stages.

Reviewer #2 (Remarks to the Author):

...I recommend that the authors determine whether altering global DNA methylation levels by over-expression of DNMT3B is directly inducing genes that lead to DNA damage/cell death or indirectly do so by specifically down-regulating the expression of Sox9. The authors could address this issue by determining whether over-expression of a Sox9 expression construct (viral or electroporated) can block the ability of ectopic DNMT3B to induce cell death and block chondrogenesis.

We thank the reviewer for this suggestion. We think that he/she is right in proposing a central role of SOX9 in this process. As we mentioned above in the replay to reviewer 1, we have initiated experiments to characterize the BMPs/DNMT3B interactions in the regulatory process. In those experiments we observed that application of BMPs up-regulated SOX9 by more than fivefold and support the point raised by the reviewer. However, we cannot keep these experiments going further, due to the closure of the University because of the coronavirus pandemic, as explained in the reply to referee 1. We think the analysis can be delayed to be addressed in a next study as we argue above. Honestly, we think that the new revision of our manuscript includes enough new data to be reconsidered in the present form.

Reviewer #3 :

Comments:

• Beyond the Methods section, the authors have made almost no reference to the species being used. The chick model should be explicitly stated in both the abstract and the introduction. Species used in experiments referred to as background should also be noted if mouse, as it remains unknown if there are species specific differences in regulation of interdigital cell death.

We solved this question in the new manuscript. We included the reference to the chick model in the abstract and along the text of the manuscript

• Authors should write out “days of incubation” at first use of i.d. abbreviation in main text, not just the methods section. Methods should have more info for the correlation of id and HH autopod stages, for those who use HH staging.

We apologize for this mistake. Now we did that at the beginning of the first paragraph of results and in the legend of the new figure supplementary 2, that shows the sequence of interdigit remodeling.

• Figure 1: the experimental design of this experiment is not clear. The authors mention sublethal doses are 1 to 4 Gy, but then sometimes use 3-4Gy and other times 1-2Gy. Why are the authors using a range of doses instead of one specific dose? Figures 1A,C,D+E should have control, non-irradiated autopod data for comparison, and ideally IHC for a range of irradiation doses.

Thank you very much for your question. The reason to employ different radiation doses was to know if there was a dose specific for undifferentiated interdigital cells but with no effects over differentiating digit cells. We identify 1 or 2 Gy as good doses to follow the degenerative events in undifferentiated interdigits mesoderm. This was supported by the foot phenotypes illustrated in the supplementary figure 1. Changes at cell level after 2Gy irradiation are now illustrated by the plot in Fig. 1 D, and in the new figures 1E, 1F, and 1G.

We have added a control as requested by the reviewer (Figure 1 A, and Suplemmentary Figure 1 A)

• SupFigure 1: A “no irradiation” control should be included as well as number and qualification of phenotypes observed. How does this data fit in with existing chick limb irradiation studies (ex. Galloway et al Nature 2009)?

We have now included the control requested by the reviewer.

Concerning the second point of the question, we must say that the observations by Galloway et al (2010) are hardly comparable with our observations. They performed their experiments in early embryonic stages trying to eliminate skeletal progenitors with distinct morphogenetic commitment. They wanted to know, if the morphogenetic fate of the progenitors (i.e. to form proximal or distal skeletal elements) was determined previously or after irradiation. Our task was to detect differences in the sensitivity of progenitors to DNA damage (DNAD), not to eliminate a specific population of progenitors. Consistent with the different tasks of both studies, our irradiation protocol was very different from that employed by Galloway et al. These authors, employed irradiation doses ranging from 3,7 Gy to up to 20 Gy. In our experiments we employed between 1 and 4 Gy to correlate the different sensitivity of cells to irradiation with the resulting phenotypes. Our findings clearly show that at lower irradiation doses (1-2 Gy) DNAD affects mainly to undifferentiated cells and limbs developed normally. At doses of 3 or 4 Gy, both undifferentiated progenitors and also early committed chondrogenic cells undergo massive DNA damage leading to precious TUNEL positivity and digit truncations. Over 4Gy none of the irradiated embryos survived to irradiation. To avoid lethality of the embryos, Galloway et al., targeted irradiation to the limb generating a shield to protect the embryo, or isolating the irradiated limb mesoderm tissue to generate a recombinant limb with an un-irradiated ectodermal cap that was next grafted into a host embryo. In summary, in the study of Galloway the task was to determine the morphogenetic basis for phocomelia at cellular level using irradiation doses that are lethal for both undifferentiated and differentiated cells.

• Figure 2: It is unclear whether this experiment includes irradiation or not? This reviewer presumes the analysis is in untreated limbs, but the first sentence of the section alludes to irradiation. Panels 2A+B should have titles, and data global methylation data should also be presented for digit sample at all stages to allow for direct comparison of digit to interdigit over the whole time course.

We apologize we weren't clearer. The reviewer is right, the data refers to control limbs. We have corrected the text to avoid that confusion. We have added a new plot (new Figure 3C), to compare digit versus interdigits methylation.

• The authors are making a case that the interdigital cells are sensitized to undergo cell death by these epigenetic marks while the digit progenitors are not. With this in mind, the IHC in Figure 3 should also include digit cells as a control.

We have included a new illustration of progenitors of the digit tip (New figure 4F)

• The data in Figure 6 is quite compelling but it begs the question of whether the same effect would be found in vivo—have the authors tried expressing these constructs via in ovo electroporation or local in vivo treatment with 5-aza? Separately, it is unclear which vector control is being used as the dotted lines in Fig 6A+B (and same comment in Fig 7).

We could not address the reviewer suggestion. We were unable to perform genetic experiments in vivo. The chick limb at advanced stages of development, is not an appropriate model to perform genetic manipulations in vivo. Often, in chicken embryos, in vivo functional assays via electroporation are performed in early organs containing a lumen (i.e. neural tube; otic placode; etc), but, in our experience, local electroporation within the dense mesenchymal tissue of the interdigits, is not efficient for gene delivery or silencing and causes important developmental alterations. As proposed by the reviewer, we attempted to implant 5-aza-beads in the interdigits, but we could not find, so far, a type of bead (heparin, acrylic, glass) able to deliver efficiently the drug in the interdigital space. We obtain in very few cases (3 out of 60 embryos) an ectopic small cartilage, but, although in our hands interdigital implantation of a single control bead has never effects, the low rate of success does not allow to be assumed as a reproducible result.

Concerning the controls, in all cases we employed cells transfected with the empty vector, that is different for gain-of-function experiments and gene silencing. We changed the legends of the figures, to clarify this question.

We finally want to thank all three reviewers for their positive comments and valuable suggestion to improve the manuscript.

Yours sincerely

Juan A Montero.

REVIEWERS' COMMENTS:

Reviewer #1 (Remarks to the Author):

The authors have fully addressed my concerns. Where appropriate, new data has been added, which substantially improved the manuscript. Congratulations to the authors!

One comment:

Fig. 3D I think the X-axis labeling ("id" for interdigit tissue) is wrong: Figure legend says it is digit tissue.

Reviewer #2 (Remarks to the Author):

This is a nicely performed study in which the authors compare the effects of ionizing radiation and implantation of a BMP-laden bead into either the digit or interdigit region of the developing limb bud. The authors document that cells in the interdigit regions (but not the chondrogenic digit regions) respond to both these perturbations with high levels of DNA damage and cell death. In addition, the authors document that DNMT1, DNMT3A, and DNMT3B are all more highly expressed in the interdigital mesenchyme in comparison to the forming digits. While over-expression of DNMT3B in micromass cultures of limb bud mesenchymal cells increased global DNA methylation, and simultaneously increased cell death and blocked chondrogenic differentiation; over-expression of an shDNMT3B had the opposite effect. The authors make the argument that relative levels of DNA methylation distinguish whether limb bud mesenchymal cells will either undergo chondrogenesis or cell death.

I think that this work is suitable for publication in Nature Communications Biology. I had previously recommended that the authors determine whether altering global DNA methylation levels by over-expression of DNMT3B is directly inducing genes that lead to DNA damage/cell death or indirectly do so by specifically down-regulating the expression of Sox9. However, as it is impossible for the authors to get into the lab to perform this experiment at the present time (due to the Covid19 crises), I recommend that this work be published in its present form. This study is a valuable contribution to the literature even without this additional experiment.